METHODS

# Fine-mapping from summary data with the "Sum of Single Effects" model

**Yuxin Zou**[1], **Peter Carbonetto**[2,3], **Gao Wang**[4]\*, **Matthew Stephens**[1,2]\*

**1** Department of Statistics, University of Chicago, Chicago, Illinois, United States of America, **2** Department of Human Genetics, University of Chicago, Chicago, Illinois, United States of America, **3** Research Computing Center, University of Chicago, Chicago, Illinois, United States of America, **4** Department of Neurology and the Gertrude. H. Sergievsky Center, Columbia University, New York, New York, United States of America

\* wang.gao@columbia.edu (GW); mstephens@uchicago.edu (MS)

**Data Availability Statement:** The genotype data used in our analyses are available from UK Biobank (https://www.ukbiobank.ac.uk). All code implementing the simulations, and the raw and compiled results generated from our simulations,

## Abstract

In recent work, Wang *et al* introduced the "Sum of Single Effects" (*SuSiE*) model, and showed that it provides a simple and efficient approach to fine-mapping genetic variants from individual-level data. Here we present new methods for fitting the *SuSiE* model to summary data, for example to single-SNP *z*-scores from an association study and linkage disequilibrium (LD) values estimated from a suitable reference panel. To develop these new methods, we first describe a simple, generic strategy for extending any individual-level data method to deal with summary data. The key idea is to replace the usual regression likelihood with an analogous likelihood based on summary data. We show that existing fine-mapping methods such as FINEMAP and CAVIAR also (implicitly) use this strategy, but in different ways, and so this provides a common framework for understanding different methods for fine-mapping. We investigate other common practical issues in fine-mapping with summary data, including problems caused by inconsistencies between the *z*-scores and LD estimates, and we develop diagnostics to identify these inconsistencies. We also present a new refinement procedure that improves model fits in some data sets, and hence improves overall reliability of the *SuSiE* fine-mapping results. Detailed evaluations of fine-mapping methods in a range of simulated data sets show that *SuSiE* applied to summary data is competitive, in both speed and accuracy, with the best available fine-mapping methods for summary data.

## Author summary

The goal of fine-mapping is to identify the genetic variants that causally affect some trait of interest. Fine-mapping is challenging because the genetic variants can be highly correlated due to a phenomenon called linkage disequilibrium (LD). The most successful current approaches to fine-mapping frame the problem as a *variable selection problem*, and here we focus on one such approach based on the "Sum of Single Effects" (*SuSiE*) model. The main contribution of this paper is to extend *SuSiE* to work with summary data, which is often accessible when the full genotype and phenotype data are not. In the process of extending *SuSiE*, we developed a new mathematical framework that helps to explain

are available at https://github.com/stephenslab/dsc_susierss, and were deposited on Zenodo (https://doi.org/10.5281/zenodo.5611713). The methods are implemented in the R package susieR, available for download at https://github.com/stephenslab/susieR, and on CRAN at https://cran.r-project.org/package=susieR.

**Funding:** MS acknowledges support from NIH NHGRI grant R01HG002585 (https://www.genome.gov) and a grant from the Gordon and Betty Moore Foundation (https://www.moore.org). GW acknowledges support from NIH NIA grant U01AG072572 (https://www.nia.nih.gov) and funding from the Thompson Family Foundation (TAME-AD; https://www.neurology.columbia.edu/research/research-programs-and-partners/thompson-family-foundation-initiative-columbia-university-tffi). The funders had no role in study design, data collection and analysis, decision to publish, or preparation of the manuscript.

**Competing interests:** The authors have declared that no competing interests exist.

existing fine-mapping methods for summary data, why they work well (or not), and under what circumstances. In simulations, we show that *SuSiE* applied to summary data is competitive with the best available fine-mapping methods for summary data. We also show how different factors such as accuracy of the LD estimates can affect the quality of the fine-mapping.

## Introduction

Fine-mapping is the process of narrowing down genetic association signals to a small number of potential causal variants [1–4], and it is an important step in the effort to understand the genetic causes of diseases [5, 6]. However, fine-mapping is a difficult problem due to the strong and complex correlation patterns ("linkage disequilibrium", or LD) that exist among nearby genetic variants. Many different methods and algorithms have been developed to tackle the fine-mapping problem [2, 7–19]. In recent work, Wang *et al* [17] introduced a new approach to fine-mapping, *SuSiE* (short for "SUm of SIngle Effects"), which has several advantages over existing approaches: it is more computationally scalable; and it provides a new, simple way to calculate "credible sets" of putative causal variants [2, 20]. However, the algorithms in [17] also have an important limitation—they require individual-level genotype and phenotype data. In contrast, many other fine-mapping methods require access only to summary data, such as *z*-scores from single-SNP association analyses and an estimate of LD from a suitable reference panel [7, 8, 11–13, 15, 16, 21]. Requiring only summary data is useful because individual-level data are often difficult to obtain, both for practical reasons, such as the need to obtain many data sets collected by many different researchers, and for reasons to do with consent and privacy. By comparison, summary data are much easier to obtain, and many publications share such summary data [22].

In this paper, we introduce new variants of *SuSiE* for performing fine-mapping from summary data; we call these variants *SuSiE-RSS* (RSS stands for "regression with summary statistics" [23].) Our work exploits the fact that (i) the multiple regression likelihood can be written in terms of a particular type of summary data, known as *sufficient statistics* (explained below), and (ii) these sufficient statistics can be approximated from the types of summary data that are commonly available (e.g., *z*-scores from single-SNP association tests and LD estimates from suitable reference panel). In the special case where the sufficient statistics themselves are available, the second approximation is unnecessary and *SuSiE-RSS* yields the same results as *SuSiE* applied to the original individual-level data; otherwise, it yields an approximation. By extending *SuSiE* to deal with widely available summary statistics, *SuSiE-RSS* greatly expands the applicability of the *SuSiE* fine-mapping approach.

Although our main goal here is to extend *SuSiE* to work with summary data, the approach we use, and the connections it exploits, are quite general, and could be used to extend other individual-level data methods to work with summary data. This general approach has two nice features. First it deals simply and automatically with non-invertible LD matrices, which arise frequently in fine-mapping. We argue, both through theory and example, that it provides a simpler and more effective solution to this issue than some existing approaches. Second, it shows how individual-level results can be obtained as a special case of summary-data analysis, by using the sufficient statistics as summary data.

By highlighting the close connection between the likelihoods for individual-level and summary data, our work generalizes results of [11], who showed a strong connection between Bayes Factors, based on specific priors, from individual-level data and summary data. Our

results highlight that this connection is fundamentally due to a close connection between the likelihoods, and so will apply whatever prior is used (and will also apply to non-Bayesian approaches that do not use a prior). By focussing on likelihoods, our analysis also helps clarify differences and connections between existing fine-mapping methods such as FINEMAP version 1.1 [12], FINEMAP version 1.2 [21] and CAVIAR [7], which can differ in both the prior and likelihood used.

Finally, we introduce several other methodological innovations for fine-mapping. Some of these innovations are not specific to *SuSiE* and could be used with other statistical methods. We describe methods for identifying "allele flips"—alleles that are (erroneously) encoded differently in the study and reference data—and other inconsistencies in the summary data. (See also [24] for related ideas.) We illustrate how a single allele flip can lead to inaccurate fine-mapping results, emphasizing the importance of careful quality control when performing fine-mapping using summary data. We also introduce a new refinement procedure for *SuSiE* that sometimes improves estimates from the original fitting procedure.

## Description of the method

We begin with some background and notation. Let $y \in \mathbb{R}^N$ denote the phenotypes of $N$ individuals in a genetic association study, and let $X \in \mathbb{R}^{N \times J}$ denote their corresponding genotypes at $J$ genetic variants (SNPs). To simplify the presentation, we assume the $y$ are quantitative and approximately normally distributed, and that both $y$ and the columns of $X$ are centered to have mean zero, which avoids the need for an intercept term in (1) [25]. We elaborate on treatment of binary and case-control phenotypes in the Discussion below.

Fine-mapping from individual-level data is usually performed by fitting the multiple linear regression model

$$y = Xb + e, \tag{1}$$

where $b = (b_1, \ldots, b_J)^\top$ is a vector of multiple regression coefficients, $e$ is an $N$-vector of error terms distributed as $e \sim \mathcal{N}_N(\mathbf{0}, \sigma^2 I_N)$, with (typically unknown) residual variance $\sigma^2 > 0$, $I_N$ is the $N \times N$ identity matrix, and $\mathcal{N}_r(\boldsymbol{\mu}, \boldsymbol{\Sigma})$ denotes the $r$-variate normal distribution with mean $\boldsymbol{\mu}$ and variance $\Sigma$.

In this multiple regression framework, the question of which SNPs are affecting $y$ becomes a problem of "variable selection"; that is, the problem of identifying which elements of $b$ are not zero. While many methods exist for variable selection in multiple regression, fine-mapping has some special features—in particular, very high correlations among some columns of $X$, and very sparse $b$—that make Bayesian methods with sparse priors a preferred approach (e.g., [7–9]). These methods specify a sparse prior for **b**, and perform inference by approximating the posterior distribution $p(b \mid X, y)$. In particular, the evidence for SNP $j$ having a non-zero effect is often summarized by the "posterior inclusion probability" (PIP),

$$\mathrm{PIP}_j \coloneqq \Pr(b_j \neq 0 \mid X, y). \tag{2}$$

### The Sum of Single Effects (*SuSiE*) model

The key idea behind *SuSiE* [17] is to write $b$ as a sum,

$$b = \sum_{l=1}^{L} b_l, \tag{3}$$

in which each vector $b_l = (b_{l1}, \ldots, b_{lJ})^\top$ is a "single effect" vector; that is, a vector with exactly

one non-zero element. The representation (3) allows that $b$ has at most $L$ non-zero elements, where $L$ is a user-specified upper bound on the number of effects. (Consider that if single-effect vectors $b_1$ and $b_2$ have a non-zero element at the same SNP $j$, $b$ will have fewer than $L$ non-zeros.)

The special case $L = 1$ corresponds to the assumption that a region has exactly one causal SNP; *i.e.*, exactly one SNP with a non-zero effect. In [17], this special case is called the "single effect regression" (SER) model. The SER is particularly convenient because posterior computations are analytically tractable [9]; consequently, despite its limitations, the SER has been widely used [2, 26–28].

For $L > 1$, Wang *et al* [17] introduced a simple model-fitting algorithm, which they called Iterative Bayesian Stepwise Selection (IBSS). In brief, IBSS iterates through the single-effect vectors $l = 1, \ldots, L$, at each iteration fitting $b_l$ while keeping the other single-effect vectors fixed. By construction, each step thus involves fitting an SER, which, as noted above, is straightforward. Wang *et al* [17] showed that IBSS can be understood as computing an approximate posterior distribution $p(b_1, \ldots, b_L \mid X, y, \sigma^2)$, and that the algorithm iteratively optimizes an objective function known as the "evidence lower bound" (ELBO).

## Summary data for fine-mapping

Motivated by the difficulties in accessing the individual-level data $X$, $y$ from most studies, researchers have developed fine-mapping approaches that work with more widely available "summary data." Here we develop methods that use various combinations of the summary data.

1. Vectors $\hat{b} = (\hat{b}_1, \ldots, \hat{b}_J)^\top$ and $\hat{s} = (\hat{s}_1, \ldots, \hat{s}_J)^\top$ containing estimates of marginal association for each SNP $j$, and corresponding standard errors, from a simple linear regression:

$$\hat{b}_j := \frac{x_j^\top y}{x_j^\top x_j}, \tag{4}$$

$$\hat{s}_j := \sqrt{\frac{(y - x_j \hat{b}_j)^\top (y - x_j \hat{b}_j)}{N x_j^\top x_j}}. \tag{5}$$

An alternative to $\hat{b}, \hat{s}$ is the vector $\hat{z} = (\hat{z}_1, \ldots, \hat{z}_J)^\top$ of $z$-scores,

$$\hat{z}_j := \hat{b}_j / \hat{s}_j. \tag{6}$$

Many studies provide $\hat{b}$ and $\hat{s}$ (see [22] for examples), and many more provide the $z$-scores, or data that can be used to compute the $z$-scores (e.g., $\hat{z}_j$ can be recovered from the $p$-value and the sign of $\hat{b}_j$ [29]). Note that it is important that all $\hat{b}, \hat{s}$ and $\hat{z}$ be computed from the same $N$ samples.

2. An estimate, $\hat{R}$, of the in-sample LD matrix, $R$, where $R$ is the $J \times J$ SNP-by-SNP sample correlation matrix,

$$R := D_{xx}^{-1/2} X^\top X D_{xx}^{-1/2} \tag{7}$$

and where $D_{xx} := \text{diag}(X^\top X)$ is a diagonal matrix that ensures the diagonal entries of $R$ are all 1. Often, the estimate $\hat{R}$ is taken to be an "out-of-sample" LD matrix—that is, the sample correlation matrix of the same $J$ SNPs in a suitable reference panel, chosen to be genetically similar to the study population, possibly with additional shrinkage or banding steps to improve accuracy [14].

3. Optionally, the sample size $N$ and the sample variance of $\boldsymbol{y}$. (Since $\boldsymbol{y}$ is centered, the sample variance of $\boldsymbol{y}$ is simply $v_y := \boldsymbol{y}^\mathsf{T}\boldsymbol{y}/N$). Knowing these quantities is obviously equivalent to knowing $\boldsymbol{y}^\mathsf{T}\boldsymbol{y}$ and $N$, so for brevity we will use the latter. These quantities are not required, but they can be helpful as we will see later.

*We caution that if the summary statistics come from a meta-analysis, the summary statistics should be computed carefully to avoid the pitfalls highlighted in [24]. Importantly, SNPs that are not analyzed in all the individual studies in the meta-analysis should not be included in the fine-mapping.*

## *SuSiE* with summary data

A key question—and the question central to this paper—is, how do we use summary data to estimate the coefficients $\boldsymbol{b}$ in a multiple linear regression (1)? And, more specifically, how do we use them to estimate the single-effect vectors $\boldsymbol{b}_1, \ldots, \boldsymbol{b}_L$ in *SuSiE* (3)? Here, we tackle these questions in two steps. First, we consider a special type of summary data, called "sufficient statistics," which contain the same information about the model parameters as the individual-level data $\boldsymbol{X}, \boldsymbol{y}$. Given such sufficient statistics, we develop an algorithm that *exactly reproduces* the results that would have been obtained by running *SuSiE* on the original data $\boldsymbol{X}, \boldsymbol{y}$. Second, we consider the case where we have access to summary data that are not sufficient statistics; these summary data can be used to approximate the sufficient statistics, and therefore approximate the results from individual-level data.

**The IBSS-ss algorithm.** The IBSS algorithm of [17] fits the *SuSiE* model to individual-level data $\boldsymbol{X}, \boldsymbol{y}$. The data enter the *SuSiE* model only through the likelihood, which from (1) is

$$\ell(\boldsymbol{b}, \sigma^2; \boldsymbol{X}, \boldsymbol{y}) = (2\pi\sigma^2)^{-N/2}\exp\left\{-\frac{1}{2\sigma^2}(\boldsymbol{y}^\mathsf{T}\boldsymbol{y} - 2\boldsymbol{b}^\mathsf{T}\boldsymbol{X}^\mathsf{T}\boldsymbol{y} + \boldsymbol{b}^\mathsf{T}\boldsymbol{X}^\mathsf{T}\boldsymbol{X}\boldsymbol{b})\right\}. \tag{8}$$

This likelihood depends on the data only through $\boldsymbol{X}^\mathsf{T}\boldsymbol{X}, \boldsymbol{X}^\mathsf{T}\boldsymbol{y}, \boldsymbol{y}^\mathsf{T}\boldsymbol{y}$ and $N$. Therefore, these quantities are *sufficient statistics*. (These sufficient statistics can be computed from other combinations of summary data, which are therefore also sufficient statistics; we discuss this point below.) Careful inspection of the IBSS algorithm in [17] confirms that it depends on the data only through these sufficient statistics. Thus, by rearranging the computations we obtain a variant of IBSS, called "IBSS-ss", that can fit the *SuSiE* model from sufficient statistics; see S1 Text.

We use IBSS($\boldsymbol{X}, \boldsymbol{y}$) to denote the result of applying the IBSS algorithm to the individual-level data, and IBSS-ss($\boldsymbol{X}^\mathsf{T}\boldsymbol{X}, \boldsymbol{X}^\mathsf{T}\boldsymbol{y}, \boldsymbol{y}^\mathsf{T}\boldsymbol{y}, N$) to denote the results of applying the IBSS-ss algorithm to the sufficient statistics. These two algorithms will give the same result,

$$\text{IBSS}(\boldsymbol{X}, \boldsymbol{y}) = \text{IBSS-ss}(\boldsymbol{X}^\mathsf{T}\boldsymbol{X}, \boldsymbol{X}^\mathsf{T}\boldsymbol{y}, \boldsymbol{y}^\mathsf{T}\boldsymbol{y}, N). \tag{9}$$

However, the computational complexity of the two approaches is different. First, computing the sufficient statistics requires computing the $J \times J$ matrix $\boldsymbol{X}^\mathsf{T}\boldsymbol{X}$, which is a non-trivial computation, requiring $O(NJ^2)$ operations. However, once this matrix has been computed, IBSS-ss requires $O(J^2)$ operations per iteration, whereas IBSS requires $O(NJ)$ operations per iteration. (The number of iterations should be the same.) Therefore, when $N \gg J$, which is often the case in fine-mapping studies, IBSS-ss will usually be faster. In practice, choosing between these workflows also depends on whether one prefers to precompute $\boldsymbol{X}^\mathsf{T}\boldsymbol{X}$, which can be done conveniently in programs such as PLINK [30] or LDstore [31].

**SuSiE with summary data: SuSiE-RSS.** In practice, sufficient statistics may not be available; in particular, when individual-level data are unavailable, the matrix $\boldsymbol{X}^\mathsf{T}\boldsymbol{X}$ is also usually unavailable. A natural approach to deal with this issue is to approximate the sufficient statistics,

then to proceed as if the sufficient statistics were available by inputting the approximate sufficient statistics to the IBSS-ss algorithm. We call this approach "*SuSiE-RSS*".

For example, let $\hat{V}_{xx}$ denote an approximation to the sample covariance $V_{xx} = \frac{1}{N}X^{\mathsf{T}}X$, and assume the other sufficient statistics $X^{\mathsf{T}}y$, $y^{\mathsf{T}}y$, $N$ are available exactly. (These are easily obtained from commonly available summary data, and $\hat{R}$; see S1 Text.) Then *SuSiE-RSS* is the result of running the IBSS-ss algorithm on the sufficient statistics but with $N\hat{V}_{xx}$ replacing $X^{\mathsf{T}}X$; that is, *SuSiE-RSS* is IBSS-ss($N\hat{V}_{xx}, X^{\mathsf{T}}y, y^{\mathsf{T}}y, N$).

In practice, we found that estimating $\sigma^2$ sometimes produced very inaccurate estimates, presumably due to inaccuracies in $\hat{V}_{xx}$ as an approximation to $V_{xx}$. (This problem did not occur when $\hat{V}_{xx} = V_{xx}$.) Therefore, when running the IBSS-ss algorithm on approximate summary statistics, we recommend to fix the residual variance, $\sigma^2 = y^{\mathsf{T}}y/N$, rather than estimate it.

**Interpretation in terms of an approximation to the likelihood.** We defined *SuSiE-RSS* as the application the IBSS-ss algorithm to the sufficient statistics or approximations to these statistics. Conceptually, this approach combines the *SuSiE* prior with an approximation to the likelihood (8).

To formalize this, we write the likelihood (8) explicitly as a function of the sufficient statistics,

$$\ell_{\text{ss}}(\boldsymbol{b}, \sigma^2; \boldsymbol{V}_{xx}, \boldsymbol{v}_{xy}, v_{yy}, N) \coloneqq (2\pi\sigma^2)^{-N/2}\exp\left\{-\frac{N}{2\sigma^2}(v_{yy} - 2\boldsymbol{b}^{\mathsf{T}}\boldsymbol{v}_{xy} + \boldsymbol{b}^{\mathsf{T}}\boldsymbol{V}_{xx}\boldsymbol{b})\right\}, \qquad (10)$$

so that

$$\ell(\boldsymbol{b}, \sigma^2; \boldsymbol{X}, \boldsymbol{y}) = \ell_{\text{ss}}\left(\boldsymbol{b}, \sigma^2; \boldsymbol{V}_{xx}, \frac{1}{N}\boldsymbol{X}^{\mathsf{T}}\boldsymbol{y}, \frac{1}{N}\boldsymbol{y}^{\mathsf{T}}\boldsymbol{y}, N\right). \qquad (11)$$

Replacing $\boldsymbol{V}_{xx}$ with an estimate $\hat{\boldsymbol{V}}_{xx}$ is therefore the same as replacing the likelihood (11) with

$$\ell_{\text{RSS}}(\boldsymbol{b}, \sigma^2) \coloneqq \ell_{\text{ss}}\left(\boldsymbol{b}, \sigma^2; \hat{\boldsymbol{V}}_{xx}, \frac{1}{N}\boldsymbol{X}^{\mathsf{T}}\boldsymbol{y}, \frac{1}{N}\boldsymbol{y}^{\mathsf{T}}\boldsymbol{y}, N\right). \qquad (12)$$

Note that when $\hat{\boldsymbol{V}}_{xx} = \boldsymbol{V}_{xx}$, the approximation is exact; that is, $\ell_{\text{RSS}}(\boldsymbol{b}, \sigma^2) = \ell(\boldsymbol{b}, \sigma^2; \boldsymbol{X}, \boldsymbol{y})$. Thus, applying *SuSiE-RSS* with $\boldsymbol{V}_{xx}$ is equivalent to using the individual-data likelihood (8), and applying it with $\hat{\boldsymbol{V}}_{xx}$ is equivalent to using the approximate likelihood (12). Finally, fixing $\sigma^2 = \frac{1}{N}\boldsymbol{y}^{\mathsf{T}}\boldsymbol{y}$ is equivalent to using the following likelihood:

$$\begin{aligned}\ell_{\text{RSS}}(\boldsymbol{b}) &\coloneqq \ell_{\text{RSS}}\left(\boldsymbol{b}, \frac{1}{N}\boldsymbol{y}^{\mathsf{T}}\boldsymbol{y}\right) \\ &= \ell_{\text{ss}}\left(\boldsymbol{b}, \frac{1}{N}\boldsymbol{y}^{\mathsf{T}}\boldsymbol{y}; \hat{\boldsymbol{V}}_{xx}, \frac{1}{N}\boldsymbol{X}^{\mathsf{T}}\boldsymbol{y}, \frac{1}{N}\boldsymbol{y}^{\mathsf{T}}\boldsymbol{y}, N\right).\end{aligned} \qquad (13)$$

**General strategy for applying regression methods to summary data.** The strategy used here to extend *SuSiE* to summary data is quite general, and could be used to extend essentially *any* likelihood-based multiple regression method for individual-level data $\boldsymbol{X}, \boldsymbol{y}$ to summary data. Operationally, this strategy would involve two steps: (i) implement an algorithm that accepts as input sufficient statistics and outputs the same result as the individual-level data; (ii) apply this algorithm to approximations of the sufficient statistics computed from (non-sufficient) summary data (optionally, fixing the residual variance to $\sigma^2 = \boldsymbol{y}^{\mathsf{T}}\boldsymbol{y}/N$). This involves replacing the exact likelihood (18) with an approximate likelihood, either (12) or (13).

**Special case when $X$, $y$ are standardized.** In genetic association studies, it is common practice to standardize both $y$ and the columns of $X$ to have unit variance—that is, $y^\mathsf{T}y = N$ and $x_j^\mathsf{T}x_j = N$ for all $j = 1, \ldots, J$—before fitting the model (1). Standardizing $X$, $y$ is commonly done in genetic association analysis and fine-mapping, and results in some simplifications that facilitates connections with existing methods, so we consider this special case in detail. (See [32, 33] for a discussion on the choice to standardize.)

When $X$, $y$ are standardized, the sufficient statistics are easily computed from the in-sample LD matrix $R$, the single-SNP $z$-scores $\hat{z}$, and the sample size, $N$:

$$X^\mathsf{T}X = NR \tag{14}$$

$$X^\mathsf{T}y = \sqrt{N}\tilde{z} \tag{15}$$

$$y^\mathsf{T}y = N, \tag{16}$$

where we define

$$\tilde{z} := D_z^{1/2}\hat{z}, \tag{17}$$

and we define $D_z$ to be the diagonal matrix in which the $j$th diagonal element is $N/(N + \hat{z}_j^2)$ [21]. Note the elements of $D_z$ have the interpretation as being one minus the estimated PVE ("Proportion of phenotypic Variance Explained"), so we refer to $\tilde{z}$ as the vector of the "PVE-adjusted $z$-scores." If all the effects are small, the estimated PVEs will be close to zero, the diagonal of $D_z$ will be close to one, and $\tilde{z} \approx \hat{z}$.

Substituting Eqs (14)–(16) into (11) gives

$$\ell(b, \sigma^2; X, y) = \ell_{ss}(b, \sigma^2; R, \tilde{z}/\sqrt{N}, 1, N). \tag{18}$$

When the in-sample LD matrix $R$ is not available, and is replaced with $\hat{R} \approx R$, the *SuSiE-RSS* likelihood (13) becomes

$$\ell_{\mathrm{RSS}}(b) = \ell_{ss}(b, 1; \hat{R}, \tilde{z}/\sqrt{N}, 1, N). \tag{19}$$

These expressions are summarized in Table 1.

**Connections with previous work.** The approach we take here is most closely connected with the approach used in FINEMAP (versions 1.2 and later) [21]. In essence, FINEMAP 1.2 uses the same likelihoods (18, 19) as we use here, but the derivations in [21] do not clearly distinguish the case where the in-sample LD matrix is available from the case where it is not. In addition, the derivations in [21] focus on Bayes Factors computed with particular priors, rather than focussing on the likelihood. Our derivations emphasize that, when the in-sample LD matrix is available, results from "summary data" should be identical to those that would have been obtained from individual-level data. Our focus on likelihoods draws attention the

**Table 1. Summary of *SuSiE* and *SuSiE-RSS*, the different data they accept, and the corresponding likelihoods.** In the "likelihood" column, $\tilde{z} := D_z^{1/2}\hat{z}$ is the vector of adjusted $z$-scores; see (17). In this summary, we assume $X$, $y$ are standardized, which is common practice in genetic association studies. Note that when *SuSiE-RSS* is applied to sufficient statistics and $\sigma^2$ is estimated (second row), the likelihood is identical to the likelihood for *SuSiE* applied the individual-level data (first row). See https://stephenslab.github.io/susieR/articles/susie_rss.html for an illustration of how these methods are invoked in the R package susieR.

| method | data type | data | $\sigma^2$ | likelihood | algorithm |
|---|---|---|---|---|---|
| *SuSiE* | individual | $X, y$ | fit | $\ell(b, \sigma^2) = \ell(b, \sigma^2; X, y)$ | IBSS |
| *SuSiE-RSS* | sufficient | $R, \hat{z}, N$ | fit | $\ell(b, \sigma^2) = \ell_{ss}(b, \sigma^2; R, \tilde{z}/\sqrt{N}, 1, N)$ | IBSS-ss |
| *SuSiE-RSS* | summary | $\hat{R}, \hat{z}, N$ | 1 | $\ell(b) = \ell_{ss}(b, 1; \hat{R}, \tilde{z}/\sqrt{N}, 1, N)$ | IBSS-ss |

generality of this strategy; it is not specific to a particular prior, nor does it require the use of Bayesian methods.

Several other previous fine-mapping methods (e.g., [7, 8, 12, 16]) are based on the following model:

$$\hat{z} \mid z, \hat{R} \sim \mathcal{N}_J(\hat{R}z, \hat{R}), \tag{20}$$

where $z = (z_1, \ldots, z_J)^\mathsf{T}$ is an unobserved vector of scaled effects, sometimes called the noncentrality parameters (NCPs),

$$z := \frac{b\sqrt{N}}{\sigma}. \tag{21}$$

(Earlier versions of *SuSiE-RSS* were also based on this model [34].) To connect our method with this approach, note that, when $\hat{R}$ is invertible, the likelihood (19) is equivalent to the likelihood for $b$ in the following model:

$$\tilde{z} \mid b, \hat{R} \sim \mathcal{N}_J(\sqrt{N}\hat{R}b, \hat{R}). \tag{22}$$

(See S1 Text for additional notes.) This model was also used in Zhu and Stephens [23], where the derivation was based on the PVE-adjusted standard errors, which gives the same PVE-adjusted $z$-scores. Model (22) is essentially the same as (20) but with the observed $z$-scores, $\hat{z}$, replaced with the PVE-adjusted $z$-scores, $\tilde{z}$. In other words, when $\hat{R}$ is invertible, these previous approaches are the same as our approach except that they use the $z$-scores, $\hat{z}$, instead of the PVE-adjusted $z$-scores, $\tilde{z}$. Thus, these previous approaches are implicitly making the approximation $X^\mathsf{T}y \approx \sqrt{N}\hat{z}$, whereas our approach uses the identity $X^\mathsf{T}y = \sqrt{N}\tilde{z}$ (Eq 15). If all effect sizes are small (*i.e.*, PVE $\approx 0$ for all SNPs), then $\tilde{z} \approx \hat{z}$, and the approximation will be close to exact; on the other hand, if the PVE is not close to zero for one or more SNPs, then the use of the PVE-adjusted $z$-scores is preferred [21]. Note that the PVE-adjusted $z$-scores require knowledge of $N$; in rare cases where $N$ is unknown, replacing $\tilde{z}$ with $\hat{z}$ may be an acceptable approximation.

**Approaches to dealing with a non-invertible LD matrix.** One complication that can arise in working directly with models (20) or (22) is that $\hat{R}$ is often not invertible. For example, if $\hat{R}$ is the sample correlation matrix from a reference panel, $\hat{R}$ will not be invertible (*i.e.*, singular) whenever the number of individuals in the panel is less than $J$, or whenever any two SNPs are in complete LD in the panel. In such cases, these models do not have a density (with respect to the Lebesgue measure). Methods using (20) have therefore required workarounds to deal with this issue. One approach is to modify ("regularize") $\hat{R}$ to be invertible by adding a small, positive constant to the diagonal [7]. In another approach, the data are transformed into a lower-dimensional space [35, 36], which is equivalent to replacing $\hat{R}^{-1}$ with its pseudoinverse (see S1 Text). Our approach is to use the likelihood (19), which circumvents these issues because the likelihood is defined whether or not $\hat{R}$ is invertible. (The likelihood is defined even if $\hat{R}$ is not positive semi-definite, but its use in that case may be problematic as the likelihood may be unbounded; see [37].) This approach has several advantages over the data transformation approach: it is simpler; it does not involve inversion or factorization of a (possibly very large) $J \times J$ matrix; and it preserves the property that results under the SER model do not depend on LD (see Results and S1 Text). Also note that this approach can be combined with modifications to $\hat{R}$, such as adding a small constant to the diagonal. The benefits of regularizing $\hat{R}$ are investigated in the experiments below.

## New refinement procedure for more accurate CSs

As noted in [17], the IBSS algorithm can sometimes converge to a poor solution (a local optimum of the ELBO). Although this is rare, it can produce misleading results when it does occur; in particular it can produce false positive CSs (*i.e.*, CSs containing only null SNPs that have zero effect). To address this issue, we developed a simple refinement procedure for escaping local optima. The procedure is heuristic, and is not guaranteed to eliminate all convergence issues, but in practice it often helps in those rare cases where the original IBSS had problems. The refinement procedure applies equally to both individual-level data and summary data.

In brief, the refinement procedure involves two steps: first, fit a *SuSiE* model by running the IBSS algorithm to convergence; next, for each CS identified from the fitted *SuSiE* model, rerun IBSS to convergence after first removing all SNPs in the CS (which forces the algorithm to seek alternative explanations for observed associations), then try to improve this fit by running IBSS to convergence again, with all SNPs. If these refinement steps improve the objective function, the new solution is accepted; otherwise, the original solution is kept. This process is repeated until the refinement steps no longer make any improvements to the objective. By construction, this refinement procedure always produces a solution whose objective is at least as good as the original IBSS solution. For full details, see S1 Text.

Because the refinement procedure reruns IBSS for each CS discovered in the initial round of model fitting, the computation increases with the number of CSs identified. In data sets with many CSs, the refinement procedure may be quite time consuming.

## Other improvements to fine-mapping with summary data

Here we introduce additional methods to improve accuracy of fine-mapping with summary data. These methods are not specific to *SuSiE* and can be used with other fine-mapping methods.

**Regularization to improve consistency of the estimated LD matrix.** Accurate fine-mapping requires $\hat{R}$ to be an accurate estimate of $R$. When $\hat{R}$ is computed from a reference panel, the reference panel should not be too small [31], and should be of similar ancestry to the study sample. Even when a suitable panel is used, there will inevitably be differences between $\hat{R}$ and $R$. A common way to improve estimation of covariance matrices is to use regularization [38], replacing $\hat{R}$ with $\hat{R}_\lambda$,

$$\hat{R}_\lambda := (1-\lambda)\hat{R}_0 + \lambda I, \tag{23}$$

where $\hat{R}_0$ is the sample correlation matrix computed from the reference panel, and $\lambda \in [0, 1]$ controls the amount of regularization. This strategy has previously been used in fine-mapping from summary data (e.g., [8, 37, 39]), but in previous work $\lambda$ was usually fixed at some arbitrarily small value, or chosen via cross-validation. Here, we estimate $\lambda$ by maximizing the likelihood under the null ($z = 0$),

$$\hat{\lambda} := \underset{\lambda \in [0,1]}{\operatorname{argmax}} \, \mathcal{N}_J(\tilde{z}; 0, (1-\lambda)\hat{R}_0 + \lambda I). \tag{24}$$

The estimated $\hat{\lambda}$ reflects the consistency between the (PVE-adjusted) *z*-scores and the LD matrix $\hat{R}_0$; if the two are consistent with one another, $\hat{\lambda}$ will be close to zero.

**Detecting and removing large inconsistencies in summary data.** Regularizing $\hat{R}$ can help to address subtle inconsistencies between $\hat{R}$ and $R$. However, regularization does not deal well with large inconsistencies in the summary data, which, in our experience, occur often. One common source of such inconsistencies is an "allele flip" in which the alleles of a SNP are

encoded one way in the study sample (used to compute $\hat{z}$) and in a different way in the reference panel (used to compute $\hat{R}$). Large inconsistencies can also arise from using $z$-scores that were obtained using different samples at different SNPs (which should be avoided by performing genotype imputation [23]). Anecdotally, we have found large inconsistencies like these often cause *SuSiE* to converge very slowly and produce misleading results, such as an unexpectedly large number of CSs, or two CSs containing SNPs that are in strong LD with each other. We have therefore developed diagnostics to help users detect such anomalous data. (We note that similar ideas were proposed in the recent paper [40].)

Under model (22), the conditional distribution of $\tilde{z}_j$ given the other PVE-adjusted $z$-scores is

$$\tilde{z}_j \mid \hat{R}, \boldsymbol{b}, \tilde{\boldsymbol{z}}_{-j} \sim \mathcal{N}((\sqrt{N}b_j - \boldsymbol{\Omega}_{j,-j}\tilde{\boldsymbol{z}}_{-j})/\Omega_{jj}, 1/\Omega_{jj}), \tag{25}$$

where $\boldsymbol{\Omega} := \hat{R}^{-1}$, $\tilde{\boldsymbol{z}}_{-j}$ denotes the vector $\tilde{\boldsymbol{z}}$ excluding $\tilde{z}_j$, and $\Omega_{j,-j}$ denotes the $j$th row of $\boldsymbol{\Omega}$ excluding $\Omega_{jj}$. This conditional distribution depends on the unknown $b_j$. However, provided that the effect of SNP $j$ is small (*i.e.*, $b_j \approx 0$), or that SNP $j$ is in strong LD with other SNPs, which implies $1/\Omega_{jj} \approx 0$, we can approximate (25) by

$$\tilde{z}_j \mid \hat{R}, \tilde{\boldsymbol{z}}_{-j} \sim \mathcal{N}(-\boldsymbol{\Omega}_{j,-j}\tilde{\boldsymbol{z}}_{-j}/\Omega_{jj}, 1/\Omega_{jj}). \tag{26}$$

This distribution has been previously used to impute $z$-scores [41], and it is also used in DENTIST [40].

An initial quality control check can be performed by plotting the observed $\tilde{z}_j$ against its conditional expectation in (26), with large deviations potentially indicating anomalous $z$-scores. Since computing these conditional expectations involves the inverse of $\hat{R}$, this matrix must be invertible. When $\hat{R}$ is not invertible, we replace $\hat{R}$ with the regularized (and invertible) matrix $\hat{R}_\lambda$ following the steps described above. Note that while we have written (25) and (26) in terms of the PVE-adjusted $z$-scores, $\tilde{\boldsymbol{z}}$, it is valid to use the same expressions for the unadjusted $z$-scores, $\hat{\boldsymbol{z}}$, so long as the effect sizes are small (DENTIST uses $z$-scores instead of the PVE-adjusted $z$-scores).

A more quantitative measure of the discordance of $\tilde{z}_j$ with its expectation under the model can be obtained by computing standardized differences between the observed and expected values,

$$t_j := \sqrt{\Omega_{jj}}(\tilde{z}_j + \boldsymbol{\Omega}_{j,-j}\tilde{\boldsymbol{z}}_{-j}/\Omega_{jj}). \tag{27}$$

SNPs $j$ with largest $t_j$ (in magnitude) are most likely to violate the model assumptions, and are therefore the top candidates for followup. When any such candidates are detected, the user should check the data pre-processing steps and fix any errors that cause inconsistencies in summary data. If there is no way to fix the errors, removing the anomalous SNPs is a possible workaround. Sometimes removing a single SNP is enough to resolve the discrepancies—for example, a single allele flip can result in inconsistent $z$-scores among many SNPs in LD with the allele-flip SNP. We have also developed a likelihood-ratio statistic based on (26) specifically for identifying allele flips; see S1 Text for a derivation of this likelihood ratio and an empirical assessment of its ability to identify allele-flip SNPs in simulations. After one or more SNPs are removed, one should consider re-running these diagnostics on the filtered summary data to search for additional inconsistencies that may have been missed in the first round. Alternatively, DENTIST provides a more automated approach to filtering out inconsistent SNPs [40].

We caution that computing these diagnostics requires inverting or factorizing a $J \times J$ matrix, and may therefore involve a large computational expense—potentially a greater expense than the fine-mapping itself—when $J$, the number of SNPs, is large.

## Verification and comparison

### Fine-mapping with inconsistent summary data and a non-invertible LD matrix: An illustration

A technical issue that arises when developing fine-mapping methods for summary data is that the LD matrix is often not invertible. Several approaches to dealing with this have been suggested including modifying the LD matrix to be invertible, transforming the data into a lower-dimensional space, or replacing the inverse with the "pseudoinverse" (see "Approaches to dealing with a non-invertible LD matrix" above). In *SuSiE-RSS*, we avoid this issue by directly approximating the likelihood, so *SuSiE-RSS* does not require the LD matrix to be invertible. We summarize the theoretical relationships between these approaches in S1 Text. Here we illustrate the practical advantage of the *SuSiE-RSS* approach in a toy example.

Consider a very simple situation with two SNPs, in strong LD with each other, with observed *z*-scores $\hat{z} = (6, 7)$. Both SNPs are significant, but the second SNP is more significant. Under the assumption that exactly one of these SNPs has an effect—which allows for exact posterior computations—the second SNP is the better candidate, and should have a higher PIP. Further, we expect the PIPs to be unaffected by LD between the SNPs (see S1 Text). However, the transformation and pseudoinverse approaches—which are used by msCAVIAR [42] and in previous fine-mapping analyses [35, 36], and are also used in DENTIST to detect inconsistencies in summary data [40]—do not guarantee that either of these properties are satisfied. For example, suppose the two SNPs are in complete LD in the reference panel, so $\hat{R}$ is a $2 \times 2$ (non-invertible) matrix with all entries equal to 1. Here, $\hat{R}$ is inconsistent with the observed $\hat{z}$ because complete LD between SNPs implies their *z*-scores should be identical. (This could happen if the LD in the reference panel used to compute $\hat{R}$ is slightly different from the LD in the association study.) The transformation approach effectively adjusts the observed data $\hat{z}$ to be consistent with the LD matrix before drawing inferences; here it would adjust $\hat{z}$ to $\hat{z} = (6.5, 6.5)$, removing the observed difference between the SNPs and forcing them to be equally significant, which seems undesirable. The pseudoinverse approach turns out to be equivalent to the transformation approach (see S1 Text), and so behaves the same way. In contrast, our approach avoids this behaviour, and correctly maintains the second SNP as the better candidate; applying *SuSiE-RSS* to this toy example yields PIPs of 0.0017 for the first SNP and 0.9983 for the second SNP, and a single CS containing the second SNP only. To reproduce this result, see the examples accompanying the `susie_rss` function in the `susieR` R package.

### Effect of allele flips on accuracy of fine-mapping: An illustration

When fine-mapping is performed using *z*-scores from a study sample and an LD matrix from a different reference sample, it is crucial that the same allele encodings are used. In our experience, "allele flips," in which different allele encoding are used in the two samples, are a common source of fine-mapping problems. Here we use a simple simulation to illustrate this problem, and the steps we have implemented to diagnose and correct the problem.

We simulated a fine-mapping data set with 1,002 SNPs, in which one out of the 1,002 SNPs was causal, and we deliberately used different allele encodings in the study sample and reference panel for a non-causal SNP (see S1 Text for more details). The causal SNP is among the

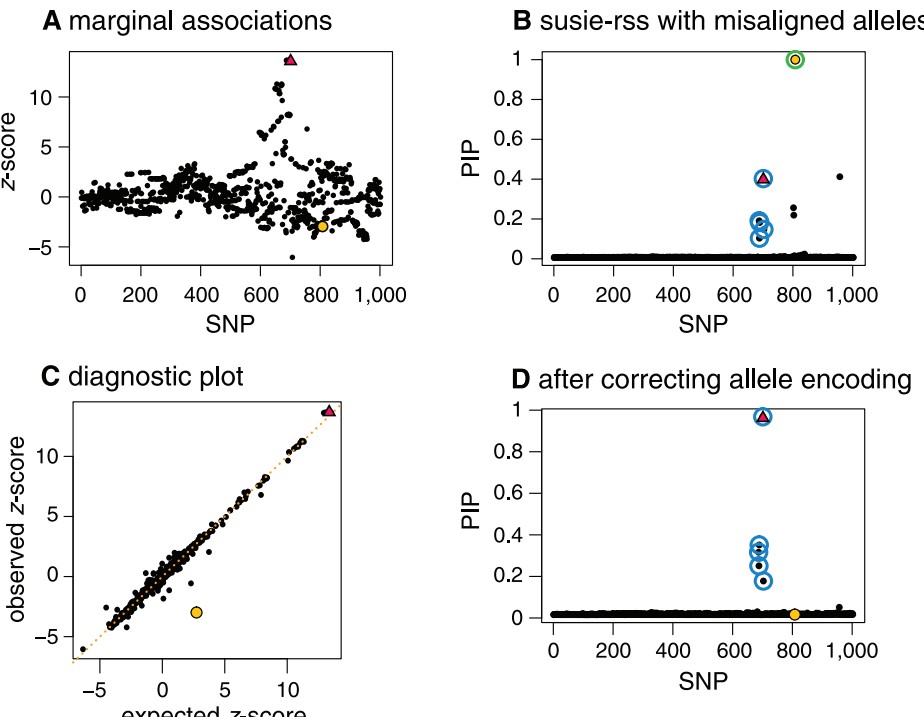

**Fig 1. Example illustrating importance of identifying and correcting allele flips in fine-mapping.** In this simulated example, one SNP (red triangle) affects the phenotype, and one SNP (yellow circle) has a different allele encoding in the study sample (the data used to compute the *z*-scores) and the reference panel (the data used to compute the LD matrix). Panel A shows the *z*-scores for all 1,002 SNPs. Panel B summarizes the results of running *SuSiE-RSS* on the summary data; *SuSiE-RSS* identifies a true positive CS (blue circles) containing the true causal SNP, and a false positive CS (green circles) that incorrectly contains the mismatched SNP. The mismatched SNP is also incorrectly estimated to have an effect on the phenotype with high probability (PIP = 1.00). The diagnostic plot (Panel C) compares the observed *z*-scores against the expected *z*-scores. In this plot, the mismatched SNP (yellow circle) shows the largest difference between observed and expected *z*-scores, and therefore appears furthest away from the diagonal. After fixing the allele encoding and recomputing the summary data, *SuSiE-RSS* identifies a single true positive CS (blue circles) containing the true-causal SNP (red triangle), and the formerly mismatched SNP is (correctly) not included in a CS (Panel D). This example is implemented as a vignette in the `susieR` package.

SNPs with the highest *z*-scores (Panel A), and *SuSiE-RSS* correctly includes this causal SNP in a CS (Panel B). However, *SuSiE-RSS* also wrongly includes the allele-flip SNP in a second CS (Panel B). This happens because the LD between the allele-flip SNP and other SNPs is incorrectly estimated. Fig 1, Panel C shows a diagnostic plot comparing each *z*-score against its expected value under model (22). The allele-flip SNP stands out as a likely outlier (yellow circle), and the likelihood ratio calculations identify this SNP as a likely allele flip: LR = $8.2 \times 10^3$ for the allele-flip SNP, whereas all the other 262 SNPs with *z*-scores greater than 2 in magnitude have likelihood ratios less than 1. (See S1 Text for a more systematic assessment of the use of these likelihood ratio for identifying allele-flip SNPs.) After correcting the allele encoding to be the same in the study and reference samples, *SuSiE-RSS* infers a single CS containing the causal SNP, and the allele-flip SNP is no longer included in a CS; see Fig 1, Panel D.

## Simulations using UK Biobank genotypes

To systematically compare our new methods with existing methods for fine-mapping, we simulated fine-mapping data sets using the UK Biobank imputed genotypes [43]. The UK Biobank

imputed genotypes are well suited to illustrate fine-mapping with summary data due to the large sample size, and the high density of available genetic variants after imputation. We randomly selected 200 regions on autosomal chromosomes for fine-mapping, such that each region contained roughly 1,000 SNPs (390 kb on average). Due to the high density of SNPs, these data sets often contain strong correlations among SNPs; on average, a data set contained 30 SNPs with correlation exceeding 0.9 with at least one other SNP, and 14 SNPs with correlations exceeding 0.99 with at least one other SNP.

For each of the 200 regions, we simulated a quantitative trait under the multiple regression model (1) with $X$ comprising genotypes of 50,000 randomly selected UK Biobank samples, and with 1, 2 or 3 causal variants explaining a total of 0.5% of variation in the trait (total PVE of 0.5%). In total, we simulated $200 \times 3 = 600$ data sets. We computed summary data from the real genotypes and synthetic phenotypes. To compare how choice of LD matrix affects fine-mapping, we used three different LD matrices: in-sample LD matrix computed from the 50,000 individuals ($R$), and two out-of-sample LD matrices computed from randomly sampled reference panels of 500 or 1,000 individuals, denoted $\hat{R}_{500}$ and $\hat{R}_{1000}$, respectively. The samples randomly chosen for each reference panel had no overlap with the study sample but were drawn from the same population, which mimicked a situation where the reference sample was well matched to the study sample.

**Refining *SuSiE* model fits improves fine-mapping performance.** Before comparing the methods, we first demonstrate the benefits of our new refinement procedure for improving *SuSiE* model fits. Fig 2 shows an example drawn from our simulations where the regular IBSS algorithm converges to a poor solution and our refinement procedure improves the solution. The example has two causal SNPs in moderate LD with one another, which have opposite effects that partially cancel out each others' marginal associations (Panel A). This example is challenging because the SNP with the strongest marginal association (SMA) is not in high LD with either causal SNP; it is in moderate LD with the first causal SNP, and low LD with the second causal SNP. Although [17] showed that the IBSS algorithm can sometimes deal well with such situations, that does not happen in this case; the IBSS algorithm yields three CSs, two of which are false positives that do not contain a causal SNP (Panel B). Applying our refinement procedure solves the problem; it yields a solution with higher objective function (ELBO), and with two CSs, each containing one of the causal SNPs (Panel C).

Although this sort of problem was not common in our simulations, it occurred often enough that the refinement procedure yielded a noticeable improvement in performance across many simulations (Fig 2, Panel D). In this plot, power and false discovery rate (FDR) are calculated as $\text{FDR} \coloneqq \frac{\text{FP}}{\text{TP+FP}}$ and $\text{power} \coloneqq \frac{\text{TP}}{\text{TP+FN}}$, where FP, TP, FN, TN denote, respectively, the number of false positives, true positives, false negatives and true negatives. In our remaining experiments, we therefore always ran *SuSiE-RSS* with refinement.

**Impact of LD accuracy on fine-mapping.** We performed simulations to compare *SuSiE-RSS* with several other fine-mapping methods for summary data: FINEMAP [12, 21], DAP-G [14, 16] and CAVIAR [7]. These methods differ in the underlying modeling assumptions, the priors used, and in the approach taken to compute posterior quantities. For these simulations, *SuSiE-RSS*, FINEMAP and DAP-G were all very fast, usually taking no more than a few seconds per data set (Table 2); by contrast, CAVIAR was much slower because it exhaustively evaluated all causal SNP configurations. Other Bayesian fine-mapping methods for summary data include PAINTOR [8], JAM [15] and CAVIARBF [11]. FINEMAP has been shown [12] to be faster and at least as accurate as PAINTOR and CAVIARBF. JAM is comparable in accuracy to FINEMAP [15] and is most beneficial when jointly fine-mapping multiple genomic regions, which we did not consider here.

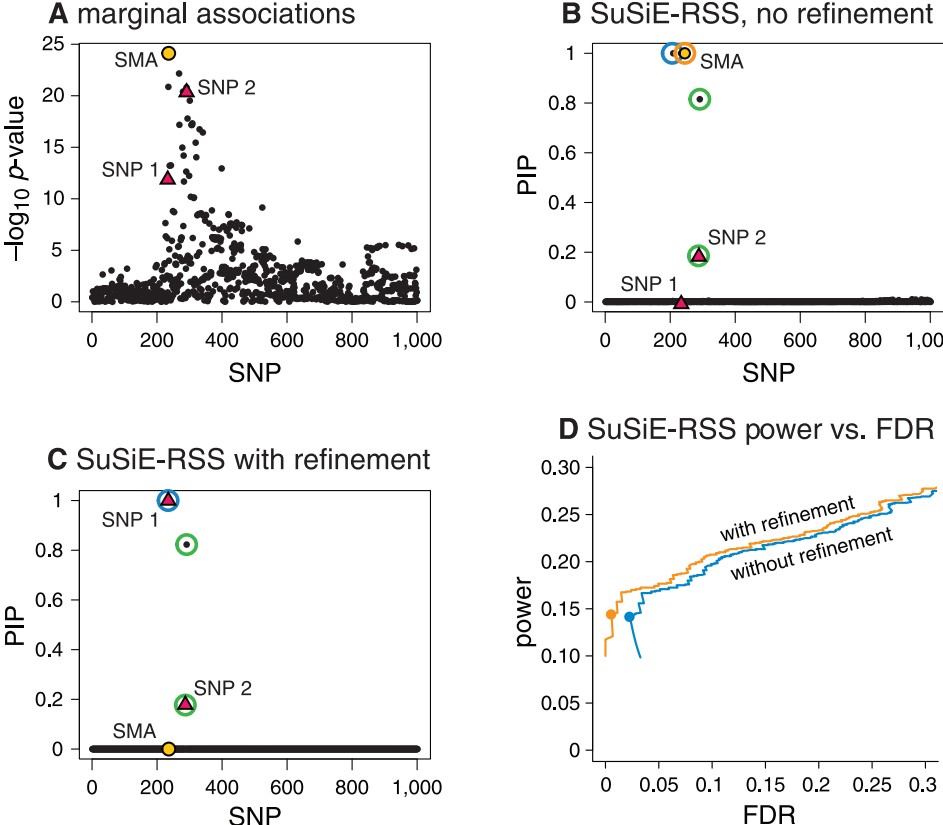

**Fig 2. Refining *SuSiE* model fits improves fine-mapping accuracy.** Panels A, B and C show a single example, drawn from our simulations, that illustrates how refining a *SuSiE-RSS* model fit improves fine-mapping accuracy. In this example, there are 1,001 candidate SNPs, and two SNPs (red triangles "SNP 1" and "SNP 2") explain variation in the simulated phenotype. The strongest marginal association (yellow circle, "SMA") is not a causal SNP. Without refinement, the IBSS-ss algorithm (applied to sufficient statistics, with estimated $\sigma^2$) returns a *SuSiE-RSS* fit identifying three 95% CSs (blue, green and orange circles); two of the CSs (blue, orange) are false positives containing no true effect SNP, one of these CSs contains the SMA (orange), and no CS includes SNP 1. After running the refinement procedure, the fit is much improved, as measured by the "evidence lower bound" (ELBO); it increases the ELBO by 19.06 (−70837.09 vs. −70818.03). The new *SuSiE-RSS* fit (Panel C) identifies two 95% CSs (blue and green circles), each containing a true causal SNP, and neither contains the SMA. Panel D summarizes the improvement in fine-mapping across all 600 simulations; it shows power and false discovery rate (FDR) for *SuSiE-RSS* with and without using the refinement procedure as the PIP threshold for reporting causal SNPs is varied from 0 to 1. (This plot is the same as a precision-recall curve after flipping the x-axis because precision $= \frac{\text{TP}}{\text{TP+FP}} = 1 - \text{FDR}$ and recall = power.) Circles are drawn at a PIP threshold of 0.95.

We compared methods based on both their posterior inclusion probabilities (PIPs) [44] and credible sets (CSs) [2, 17]. These quantities have different advantages. PIPs have the advantage that they are returned by most methods, and can be used to assess familiar quantities such as power and false discovery rates. CSs have the advantage that, when the data support multiple causal signals, the multiple causal signals is explicitly reflected in the number of CSs reported. Uncertainty in which SNP is causal is reflected in the size of a CS.

First, we assessed the performance of summary-data methods using the in-sample LD matrix. With an in-sample LD matrix, *SuSiE-RSS* applied to sufficient statistics (with estimated $\sigma^2$) will produce the same results as *SuSiE* on the individual-level data, so we did not include *SuSiE* in this comparison. The results show that *SuSiE-RSS*, FINEMAP and DAP-G have very similar performance, as measured by both PIPs (Fig 3) and CSs ("in-sample LD" columns in

**Table 2. Runtimes on simulated data sets with in-sample LD matrix.** Average runtimes are taken over 600 simulations. All runtimes are in seconds. All runtimes include the time taken to read the data and write the results to files.

| method | min. | average | max. |
|---|---|---|---|
| *SuSiE-RSS*, estimated σ, no refinement | 0.65 | 1.33 | 18.89 |
| *SuSiE-RSS*, estimated σ, with refinement | 1.62 | 5.50 | 72.57 |
| *SuSiE-RSS*, fixed σ, no refinement | 0.40 | 1.40 | 18.61 |
| *SuSiE-RSS*, fixed σ, with refinement | 1.44 | 4.81 | 62.34 |
| *SuSiE-RSS*, fixed σ, with refinement, $L$ = true | 0.37 | 1.52 | 4.95 |
| DAP-G | 0.66 | 5.70 | 371.76 |
| FINEMAP | 1.67 | 16.11 | 39.27 |
| FINEMAP, $L$ = true | 1.00 | 12.92 | 42.93 |
| CAVIAR, $L$ = true | 3.54 | 1,516.91 | 4,831.95 |

Fig 4). Further, all four methods produced CSs whose coverage was close to the target level of 95% (Panel A in Fig 4). The main difference between the methods is that DAP-G produced some "high confidence" (high PIP) false positives, which hindered its ability to produce very low FDR values. Both *SuSiE-RSS* and FINEMAP require the user to specify an upper bound on the number of causal SNPs $L$. Setting this upper bound to the true value ("$L$ = true" in the figures) only slightly improved their performance, demonstrating that, with an in-sample LD matrix, these methods are robust to overstating this bound. We also compared the sufficient-data (estimated $\sigma^2$) and summary-data (fixed $\sigma^2$) variants of *SuSiE-RSS* (see Table 1). The performance of the two variants was very similar, likely owing to the fact that the PVE was close to zero in all simulations, and so $\sigma^2 = 1$ was not far from the truth. CAVIAR performed notably less well than the other methods for the PIP computations. (Note the CSs computed by

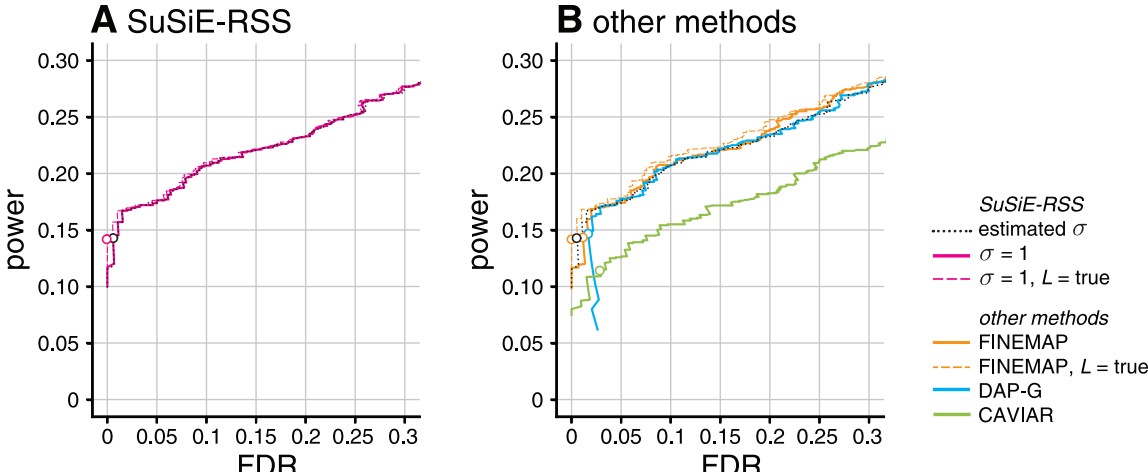

**Fig 3. Discovery of causal SNPs using posterior inclusion probabilities—in-sample LD.** Each curve shows power vs. FDR in identifying causal SNPs when the method (*SuSiE-RSS*, FINEMAP, DAP-G or CAVIAR) was provided with the in-sample LD matrix. FDR and power are calculated from 600 simulations as the PIP threshold is varied from 0 to 1. Open circles are drawn at a PIP threshold of 0.95. Two variants of FINEMAP and three variants of *SuSiE-RSS* are also compared: when $L$, the maximum number of estimated causal SNPs, is the true number of causal SNPs, or larger than the true number; and, for *SuSiE-RSS* only, when the residual variance $\sigma^2$ is estimated ("sufficient data") or fixed to 1 ("summary data"); see Table 1. The results for *SuSiE-RSS* with estimated $\sigma^2$ is shown in both A and B to aid in comparing results. Note that power and FDR are virtually identical for all three variants of *SuSiE-RSS* so the three curves almost completely overlap in Panel A.

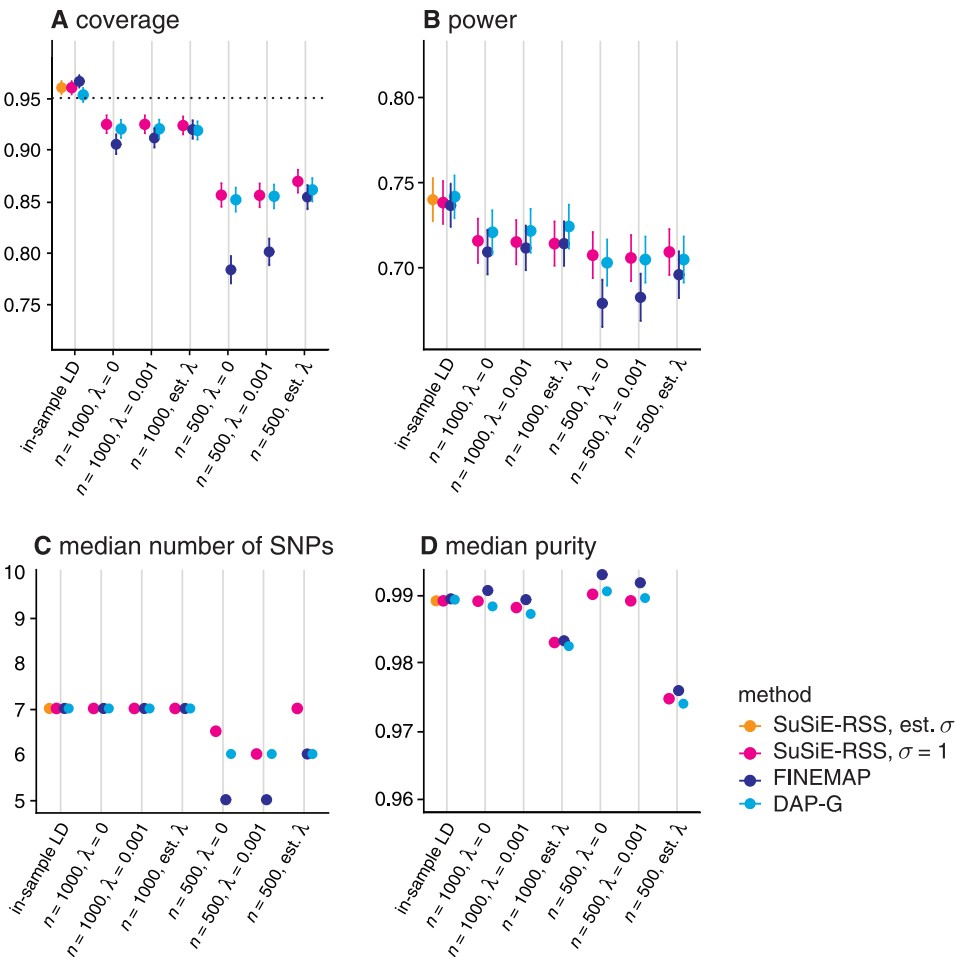

**Fig 4. Assessment of 95% credible sets from *SuSiE-RSS*, FINEMAP and DAP-G with different LD estimates, and different LD regularization methods.** For in-sample LD, two variants of *SuSiE-RSS* were also compared (see Table 1): when the residual variance $\sigma^2$ was estimated ("sufficient data"), or fixed to 1 ("summary data"). We evaluate the estimated CSs using the following metrics: (A) *coverage*, the proportion of CSs that contain a true causal SNP; (B) *power*, the proportion of true causal SNPs included in a CS; (C) median number of SNPs in each CS; and (D) *median purity*, where "purity" is defined as the smallest absolute correlation among all pairs of SNPs within a CS. These statistics are taken as the mean (A, B) or median (C, D) over all simulations; error bars in A and B show two times the standard error. The target coverage of 95% is shown as a dotted horizontal line in Panel A. Following [17], we discarded all CSs with purity less than 0.5.

CAVIAR are defined differently from CSs computed by other methods, so we did not include CAVIAR in Fig 4.)

Next, we compared the summary data methods using different out-of-sample LD matrices, again using *SuSiE-RSS* with in-sample LD (and estimated $\sigma^2$) as a benchmark. For each method, we computed out-of-sample LD matrices using two different panel sizes ($n$ = 500, 1000) and three different values for the regularization parameter, $\lambda$ (no regularization, $\lambda = 0$; weak regularization, $\lambda = 0.001$; and $\lambda$ estimated from the data). As might be expected, the performance of *SuSiE-RSS*, FINEMAP and DAP-G all degraded with out-of-sample LD compared with in-sample LD; see Figs 4 and 5. Notably, the CSs no longer met the 95% target coverage (Panel A in Fig 4). In all cases, performance was notably worse with the smaller reference panel, which highlights the importance of using a sufficiently large reference panel [31]. Regarding regularization, *SuSiE-RSS* and DAP-G performed similarly at all levels of

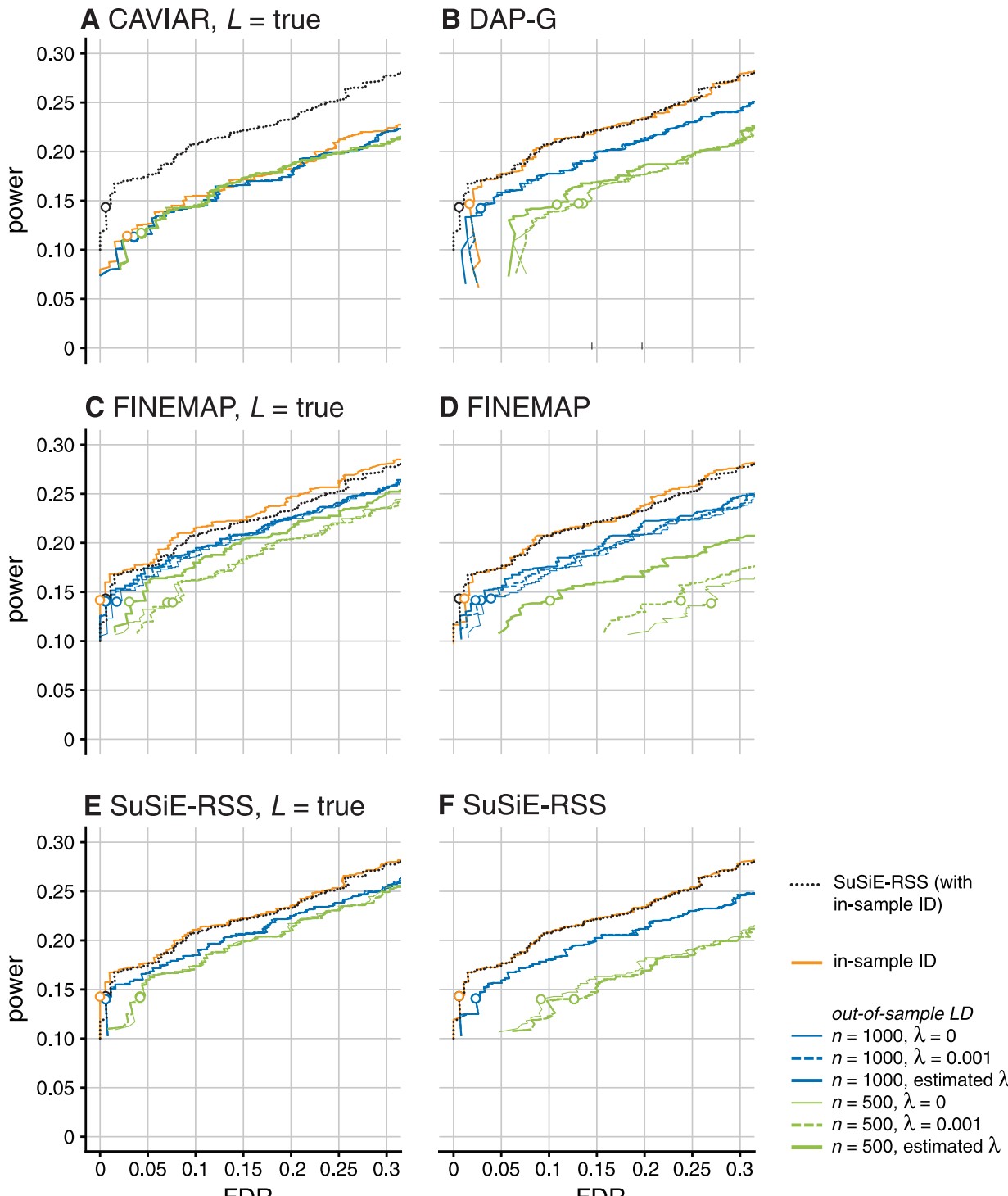

**Fig 5. Discovery of causal SNPs using posterior inclusion probabilities—Out-of-sample LD.** Plots compare power vs. FDR for fine-mapping methods with different LD matrices, across all 600 simulations, as the PIP threshold is varied from 0 to 1. Open circles indicate results at PIP threshold of 0.95. Each plot compares performance of one method (CAVIAR, DAP-G, FINEMAP or *SuSiE-RSS*) when provided with different LD estimates: in-sample ($\hat{R} = R$), or out-of-sample LD from a reference panel with either 1,000 samples ($\hat{R} = \hat{R}_{1000}$) or 500 samples ($\hat{R} = \hat{R}_{500}$). For out-of-sample LD, different levels of the regularization parameter λ are also compared: λ = 0; λ = 0.001; and estimated λ. Panels C–F show results for two variants of FINEMAP and *SuSiE-RSS*: in Panels C and E, the maximum number of causal SNPs, *L*, is set to the true value ("*L* = true"); in Panels D and F, *L* is set larger than the true value (*L* = 5 for FINEMAP; *L* = 10 for *SuSiE-RSS*). In each panel, the dotted black line shows the results from *SuSiE-RSS* with in-sample LD and estimated $\sigma^2$, which provides a baseline for comparison (note that all the other *SuSiE-RSS*

results were generated by fixing $\sigma^2$ to 1, which is the recommended setting for out-of-sample LD; see Table 1). Some power vs. FDR curves may not be visible in the plots because they overlap almost completely with another curve, such as some of the *SuSiE-RSS* results at different LD regularization levels.

regularization, and so do not appear to require regularization; in contrast, FINEMAP required regularization with an estimated λ to compete with *SuSiE-RSS* and DAP-G. Since estimating λ is somewhat computationally burdensome, *SuSiE-RSS* and DAP-G have an advantage in this situation. All three methods benefited more from increasing the size of the reference panel than from regularization, again emphasizing the importance of sufficiently large reference panels. Interestingly, CAVIAR's performance was relatively insensitive to choice of LD matrix; however the other methods clearly outperformed CAVIAR with the larger ($n = 1, 000$) reference panel.

The fine-mapping results with out-of-sample LD matrix also expose another interesting result: if FINEMAP and *SuSiE-RSS* are provided with the true number of causal SNPs ($L =$ true), their results improve (Fig 5, Panels C vs. D, Panels E vs. F). This improvement is particularly noticeable for the small reference panel. We interpret this result as indicating a tendency of these methods to react to misspecification of the LD matrix by sometimes including additional (false positive) signals. Specifying the true $L$ reduces their tendency to do this because it limits the number of signals that can be included. This suggests that restricting the number of causal SNPs, $L$, may make fine-mapping results more robust to misspecification of the LD matrix, even for methods that are robust to overstating $L$ when the LD matrix is accurate. Priors or penalties that favor smaller $L$ may also help. Indeed, when none of the methods are provided with information about the true number of causal SNPs, DAP-G slightly outperforms FINEMAP and *SuSiE-RSS*, possibly reflecting a tendency for DAP-G to favour models with smaller numbers of causal SNPs (either due to the differences in prior or differences in approximate posterior inference). Further study of this issue may lead to methods that are more robust to misspecified LD.

**Fine-mapping causal SNPs with larger effects.** Above, we evaluated the performance of fine-mapping methods in simulations when the simulated effects of the causal SNPs were small (total PVE of 0.5%). This was intended to mimic the typical situation encountered in genome-wide association studies [45, 46]. Here we scrutinize the performance of fine-mapping methods when the effects of the causal SNPs are much larger, which might be more representative of the situation in expression quantitative trait loci (eQTL) studies [47–49]. FINEMAP and *SuSiE*—and therefore *SuSiE-RSS* with sufficient statistics—are expected to perform well in this setting [17, 21], but, as mentioned above, some summary-data methods make the (implicit) assumption that the effects are small (see "Connections with previous work"), and this assumption may affect performance in settings where this assumption is violated.

To assess the ability of the fine-mapping methods to identify causal SNPs with larger effects, we performed an additional set of simulations, again using the UK Biobank genotypes, except that here we simulated the 1–3 causal variants so that they explained, in total, a much larger proportion of variance in the trait (PVE of 10% and 30%). To evaluate these methods at roughly the same level of difficulty (*i.e.*, power), we simulated these fine-mapping data sets with much smaller sample sizes, $N = 2, 500$ and $N = 800$, respectively (the out-of-sample LD matrix was calculated using 1,000 samples).

The results of these high-PVE simulations are summarized in Fig 6. As expected, *SuSiE-RSS* with in-sample LD matrix performed consistently better than the other methods, which use an out-of-sample LD matrix, and therefore provides a baseline against which other methods can

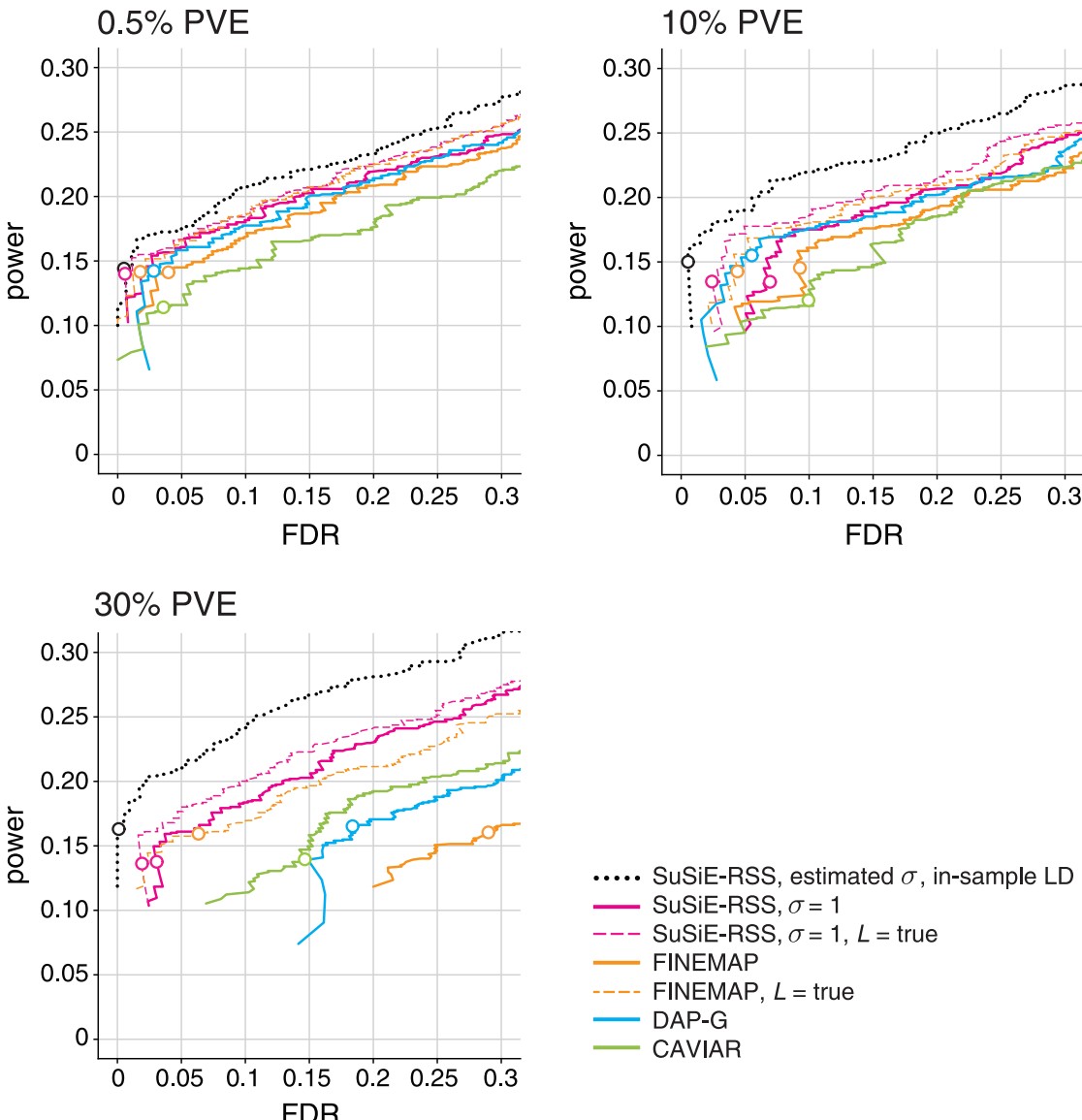

**Fig 6. Discovery of causal SNPs using posterior inclusion probabilities—Out-of-sample LD and larger effects.** Each curve shows power vs. FDR for identifying causal SNPs with different effect sizes (total PVE of 0.5%, 10% and 30%). Each panel summarizes results from 600 simulations; FDR and power are calculated from the 600 simulations as the PIP threshold is varied from 0 to 1. Open circles depict power and FDR at a PIP threshold of 0.95. In addition to comparing different methods (*SuSiE-RSS*. FINEMAP, DAP-G, CAVIAR), two variants of FINEMAP and *SuSiE-RSS* are also compared: when *L*, the maximum number of estimated causal SNPs, is set to the true number of causal SNPs; and when *L* is larger than the true number. *SuSiE-RSS* with estimated residual variance $\sigma^2$ and in-sample LD (dotted black line) is shown as a "best case" method against which other methods can be compared. All other methods are given an out-of-sample LD matrix computed from a reference panel with 1,000 samples, and with no regularization ($\lambda = 0$). The simulation results for 0.5% PVE (top-left panel) are the same as the results shown in previous plots (Figs 3 and 5), but presented differently here to facilitate comparison with the results of the higher-PVE simulations.

be compared. Overall, increasing PVE tended to increase variation in performance among the methods. In all PVE settings, *SuSiE-RSS* with out-of-sample LD was among the top performers, and it most clearly outperformed other methods in the highest PVE setting (30% PVE), where all of FINEMAP, DAP-G, and CAVIAR showed a notable decrease in performance. For DAP-G and CAVIAR, this decrease in performance was expected due to their implicit

modeling assumption that the effect sizes are small. For FINEMAP, this drop in performance was unexpected since FINEMAP also uses the PVE-adjusted $z$-scores to account for larger effects. Although this situation is unusual in fine-mapping studies—that is, it is unusual for a handful of SNPs to explain such a large proportion of variance in the trait—we examined these FINEMAP results more closely to understand why this was happening. (We also prepared a detailed working example illustrating this result; see https://stephenslab.github.io/finemap/large_effect.html.) We confirmed that this performance drop only occurred with an out-of-sample LD matrix; with an in-sample LD matrix, FINEMAP's performance was very similar to *SuSiE-RSS*'s with an in-sample LD matrix (results not shown). A partial explanation for the much worse performance with out-of-sample LD was that FINEMAP often overestimated the number of causal SNPs; in 17% of the simulations, FINEMAP assigned highest probability to configurations with more causal SNPs than the true number. By contrast, *SuSiE-RSS* overestimated the number of causal SNPs (*i.e.*, the number of CSs) in only 1% of the simulations. Fortunately, in settings where causal SNPs might have larger effects, FINEMAP's performance can be greatly improved by telling it the true number of causal SNPs ("$L$ = true"), which is consistent with our earlier finding that restricting $L$ in *SuSiE-RSS* and FINEMAP can improve fine-mapping with an out-of-sample LD matrix.

## Discussion

We have presented extensions of the *SuSiE* fine-mapping method to accommodate summary data, with a focus on marginal $z$-scores and an out-of-sample LD matrix computed from a reference panel. Our approach provides a generic template for how to extend any full-data regression method to analyze summary data: develop a full-data algorithm that works with sufficient statistics, then apply this algorithm directly to summary data. Although it is simple, as far as we are aware this generic template is novel, and it avoids the need for any special treatment of non-invertible LD matrices.

In simulations, we found that our new method, *SuSiE-RSS*, is competitive in both accuracy and computational cost with the best available methods for fine-mapping from summary data, DAP-G and FINEMAP. Whatever method is used, our results underscore the importance of accurately computing out-of-sample LD from an appropriate and large reference panel (see also [31]). Indeed, for the best performing methods, performance depended more on choice of LD matrix than on choice of method. We also emphasize the importance of computing $z$-scores at different SNPs from the exact same samples, using genotype imputation if necessary [50]. It is also important to ensure that alleles are consistently encoded in study and reference samples.

Although our derivations and simulations focused on $z$-scores computed from quantitative traits with a simple linear regression, in practice it is common to apply summary-data fine-mapping methods to $z$-scores computed in other ways, e.g., using logistic regression on a binary or case-control trait, or using linear mixed models to deal with population stratification and relatedness. The multivariate normal assumption on $z$-scores, which underlies all the methods considered here, should also apply to these settings, although as far as we are aware theoretical derivation of the precise form (20) is lacking in these settings (although see [12, 51, 52]). Since the model (20) is already only an approximation, one might expect that the additional effect of such issues might be small, particularly compared with the effect of allele flips or small reference panels. Nonetheless, since our simulations show that model misspecification can hurt performance of existing methods, further research to improve robustness of fine-mapping methods to model misspecification would be welcome.

## Supporting information

**S1 Fig. Likelihood ratio for detecting allele flips.** These plots summarize the likelihood ratios $LR_j$ for SNPs $j$ in simulated fine-mapping data sets, separately for allele-flip SNPs with an effect (top row, right-hand side), without an effect on the trait (top row, left-hand side), and for SNPs without a flipped allele that affect the trait (middle row, right-hand side) and do not affect the trait (middle row, left-hand side). The two histograms in the bottom row show likelihood ratios after restricting to SNPs with $z$-scores greater than 2 in magnitude. The bar heights in the histograms in the middle and bottom rows are drawn on the logarithmic scale to better visualize the smaller numbers of SNPs with likelihood ratios greater than 1 (*i.e.*, $\log LR_j > 0$). (PDF)

**S1 Text. Detailed methods.** More description of the methods, including: the single effect regression (SER) model with summary statistics; the IBSS-ss algorithm; computing the sufficient statistics; approaches to dealing with a non-invertible LD matrix; estimation of $\lambda$ in the regularized LD matrix; likelihood ratio for detecting allele flips; *SuSiE* refinement procedure; detailed calculations for toy example; and more details on the UK Biobank simulations. (PDF)

## Acknowledgments

We thank Kaiqian Zhang for her contributions to the development and testing of the `susieR` package. We thank the University of Chicago Research Computing Center and the Center for Research Informatics for providing high-performance computing resources used to run the numerical experiments. This research has been conducted using data from UK Biobank, a major biomedical database (UK Biobank Application Number 27386).

## Author Contributions

**Conceptualization:** Matthew Stephens.

**Data curation:** Yuxin Zou, Peter Carbonetto.

**Formal analysis:** Yuxin Zou, Peter Carbonetto, Gao Wang, Matthew Stephens.

**Funding acquisition:** Matthew Stephens.

**Investigation:** Yuxin Zou, Peter Carbonetto, Matthew Stephens.

**Methodology:** Yuxin Zou, Peter Carbonetto, Gao Wang, Matthew Stephens.

**Project administration:** Peter Carbonetto, Matthew Stephens.

**Software:** Yuxin Zou, Peter Carbonetto, Gao Wang, Matthew Stephens.

**Supervision:** Peter Carbonetto, Matthew Stephens.

**Validation:** Yuxin Zou, Matthew Stephens.

**Visualization:** Yuxin Zou, Peter Carbonetto.

**Writing – original draft:** Yuxin Zou, Peter Carbonetto, Gao Wang, Matthew Stephens.

**Writing – review & editing:** Yuxin Zou, Peter Carbonetto, Gao Wang, Matthew Stephens.

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
