## [Decision Letter · Decision Letter 0]

17 Jan 2022

Dear Dr Carbonetto,

Thank you very much for submitting your Methods manuscript entitled 'Fine-mapping from summary data with the "Sum of Single Effects" model' to PLOS Genetics. We apologize with the delay in our decision - the final reviewer report was received only now, and as there were some differences of opinion among the reviewers it was necessary to wait for the third review. 

All three reviewers found the manuscript well-written and technically sound. While Reviewers 1 and 3 were quite favorable about this work, Reviewer 2 raised issues regarding novelty and impact of the proposed method relative to other published fine-mapping methods. Based on this feedback, we have opted for a "major revision" decision, we feel that enough serious points have been raised to require substantial revision but we are not prescriptive about what specific revisions are required to respond to the reviewer concerns.  We ask you to address the first three points raised by reviewer 2 in their main paragraph, about importance and similarity to other methods, but the fourth point need not be addressed, and similarly the first "major comment" about hat notation perhaps just needs a small clarification.  The second "major comment" does need some response as well as the minor comments.

We hope that you will decide to revise the manuscript for further consideration here.  Please also submit a detailed list of your responses to the review comments and a description of the changes you have made in the manuscript.

If you decide to revise the manuscript for further consideration at PLOS Genetics, please aim to resubmit within the next 60 days, unless it will take extra time to address the concerns of the reviewers, in which case we would appreciate an expected resubmission date by email to plosgenetics@plos.org.

[LINK]

We are sorry that we cannot be more positive about your manuscript at this stage. Please do not hesitate to contact us if you have any concerns or questions.

Yours sincerely,

Michael P. Epstein

Associate Editor

PLOS Genetics

David Balding

Section Editor: Methods

PLOS Genetics

Reviewer's **Comments to the Authors:**

Reviewer #1: Review: Fine-mapping from summary data with the “Sum of Single Effects” model

Overview: Zou et al present their work that extends the sum of single effects (SuSiE) bayesian sparse model for fine-mapping to operate directly on GWAS summary statistics. To do so they describe the sufficient statistics for fine-mapping, which happens to be marginal effect-size estimates and linkage disequilibrium information (LD). They demonstrate that this setting can accommodate either effect-size estimates along with standard errors, or z-score statistics, which are much more commonly released in GWAS data. In addition to this extension, Zou et al describe approaches to perform inference in a low-rank setting, which is commonly the case when estimating LD from reference population data (e.g., 1000G). Lastly, Zou et al also describe approaches to QC summary data under model assumptions and identify SNPs with improperly labeled encoding (i.e. “allele flips”). The manuscript presents simulated data results to support their claims, is well written, and easy to follow. SuSiE has quickly become a standard approach for fine-mapping in recent years and I was excited to see the authors describe a robust summary-based version. I have some comments below.

Major Comments:

1. The main exposition of the SuSiE-suff and SuSiE-RSS-Z approaches is very well done, justified by empirical results and I have no major comments to include regarding the primary fine-mapping procedure.

Minor Comments:

1. The authors present a nice strategy to QC allele flips in summary data using a likelihood ratio test under a mixture model. Their approach is well described, and the authors provide some guidelines on how to use this in practice. The authors are clear in that this procedure should not be considered as an automated approach to QC and should be done interactively. Given that, I would still appreciate to see some QQ plot (or something similar) to see how well the fitted LRTs behave under the null and its power under the alternate. Results in this setting could further justify the interactive nature for this tool.

Reviewer #2: The paper presents a version of the fine-mapping model SuSiE that is applicable to summary data (z-scores and LD-matrix) while the previously published SuSiE model was applicable to individual level data (genotypes and phenotypes). Additionally, paper considers three topics related to practical issues of fine-mapping: (1) detecting inconsistencies between z-scores and LD estimates, (2) regularizing the estimated LD-matrix, and (3) a computational refinement procedure of SuSiE algorithm. As examples the study uses simulated phenotypes on UK biobank genetic data.

Paper is well-written and references to existing work are appropriate. Since SuSiE is a key method for fine-mapping, its implementation applicable to summary data is an important contribution to the field and it is/will be used widely. However, when considering the guidelines for publications in this forum, it is less clear what in this paper presented “a new way of approaching a biological or biomedical question, or a substantive advance over existing approaches”. First, there are well-established fine-mapping methods (such as FINEMAP and DAP-G) that work with the same idea (replace X^T y by scaled effect estimates and X^T X by scaled LD-matrix in linear model likelihood) which produce similar results as SuSiE. Second, the proposed detection of inconsistencies between z-scores and LD-matrix seems similar to recently published DENTIST method (NATURE COMMUNICATIONS, 2021 12:7117), which is also stated in this paper. Third, while, to my knowledge, these kind of LD-matrix regularization results have not been presented before in fine-mapping context, and therefore they are interesting, the regularization approach does not seem that important in practice. Fourth, the computational refinement procedure of SuSiE algorithm is a technical fix to SuSiE algorithm rather than a considerable improvement over the existing methods.

Major comments:

Hat-notation for b and z.

I find it confusing that paper defines vector b as MULTIPLE regression coefficient but uses b^ as an estimate for coefficients from SIMPLE regression. Thus, in this paper, b^ is not an estimate of b. This is against common statistical notation and therefore likely causes confusion for a very large group of readers. I would suggest using the standard notation where hat denotes an estimate of the very parameter on which the hat is put.

Same comment about z-scores.

Assumption about effects being very small.

Line 172 states that model RSS-Z is valid only when all non-zero effects are very small. While this indeed is the most common case, there are also loci that explain several percentages of variance and typically these are highly interesting loci for fine-mapping with multiple causal variants in them. Does SuSiE model handle these loci correctly when applied to individual level data? And what about summary data? If you ran your simulated examples with variance explained set to 30% instead of 0.5% what would happen with each method? Please include in the manuscript some clear example or statement about this. (At least FINEMAP should handle appropriately such cases https://doi.org/10.1101/318618 )

Minor comments:

l.87 “tractible” -> “tractable”

l.94 “approximate posterior of b_1,…b_L” : Doesn’t the algorithm converge to only one of the L! symmetrical modes of the posterior rather than to the actual posterior?

l.121 says that sufficient statistics contain “exactly the same information as individual level data”. More precisely they contain same information about the parameters of a particular model considered here. But they don’t, in general, contain “exactly the same information” as full data.

l.122 The statistics mentioned are sufficient statistics for the parameters of the SuSiE model. (They are not any general “sufficient statistics” of these data.)

l.233 It is unclear what "refine" means here? Is it “rerun until convergence starting from the current state”?

l.289 Remove extra “care”.

l.373. Say that samples have self-reported their ethnicity as white British. (If that is indeed the case.)

Figure 2 legend defines “power” and “FDR” that are then used also in other Figures. Would be better to define these in text once and then use in all Figures.

l.497 Benner et al. 2016 (FINEMAP paper) suggests how to interpret the linear model parameters to account for properties of binary data such as case-control ratio.

l.669 “then all R_jj’ do not appear in the likelihood” Do you mean “then none of R_jj’ appear in the likelihood”?

l.677-678 “rows and columns in \\gamma”. Do you mean the square submatrix of dimension |\\gamma| of R formed by subrows and subcolumns corresponding to elements in \\gamma?

Similarly for -\\gamma.

l. 712. The fact that no LD is needed to do proper inference is also true for binary traits as stated already by Maller et al. (2012). See Supp Text p. 56-57 of Maller et al. (2012) Nature Genetics 44,1294–1301.

l. 773. Hard to believe that the above mentioned criteria result in exactly 50,000 samples. Make clear how you end up with exactly 50,000?

Reviewer #3: This manuscript describes the extension of the recently proposed SuSiE model to summary data (z scores and correlation matrix), extending its applicability to fine mapping for genetic summary data.

The manuscript is clearly written, and the mathematical exposition is careful and sufficiently detailed to follow.

Inconsistency between summary estimates and the LD matrix is an important problem which can produce misleading results and slow convergence. A good section of the manuscript is dedicated to dealing with these, through regulation of the LD matrix, including estimating the regularisation parameter lambda, which I think is novel. Detecting inconsistencies though is a thorny problem, and a new method is proposed for those. However, the computational complexity is high, perhaps higher than running fitting the SuSiE model itself. And so some guidance about when this should be considered would be useful. Are there diagnostics from the SuSiE output that indicate something could be awry and suggest it would be worth running the discrepancy detection? For example, in my own experience finding multiple credible sets (up to 10) often containing only one SNP, or containing SNPs with no marginal evidence for association has indicated issues with the data.

Overall, this is an important contribution, extending the use of the new SuSiE approach to summary data. The approach is already in widespread use in the statistical genetics and bioinformatics communities, so the exposition of the underlying mathematics and associated comments on how the approach should be applied is very timely.

Minor comments:

In Fig 4, I would like to compare SuSiE-RSS to other approaches, but as each approach has its own subplot and there are no grid lines, this is very hard to do. Could the results be faceted data (LD sample size, lambda etc) or grid lines added? The dotted black line is there in each plot, but is not enough for me to decide whether the green lines in D are above/below the green lines in F, for example.

**Have all data underlying the figures and results presented in the manuscript been provided?**

Reviewer #1: Yes

Reviewer #2: Yes

Reviewer #3: Yes

PLOS authors have the option to publish the peer review history of their article (what does this mean?). If published, this will include your full peer review and any attached files.

Reviewer #1: No

Reviewer #2: No

Reviewer #3: No

---

## [Decision Letter · Decision Letter 1]

23 May 2022

Dear Dr Carbonetto,

Thank you very much for submitting your Methods entitled 'Fine-mapping from summary data with the "Sum of Single Effects" model' to PLOS Genetics.

The manuscript was fully evaluated at the editorial level and by three independent peer reviewers. While two reviewers were fully satisfied with this revision, one reviewer requested some additional clarification regarding some of the new simulations that were performed. Once you address these additional questions, our intent would then be to accept this work. 

[LINK]

Yours sincerely,

Michael P. Epstein

Associate Editor

PLOS Genetics

David Balding

Section Editor: Methods

PLOS Genetics

Reviewer's **Comments to the Authors:**

Reviewer #1: The authors have addressed my previous concerns.

Reviewer #2: I thank the Authors for their additional work related to adopting adjusted z-scores and running high PVE simulations and for their detailed answers to previous comments.

These new results pose one more question, namely why FINEMAP performs so much worse than SuSiE with increasing PVE when both should handle the situation through the same adjusted z-scores and when Benner et al. 2018 did not observe problems with FINEMAP in high PVE cases. To understand this, it would be useful to know whether the FINEMAP problem in high PVE cases appear only with approximate LD matrix (currently shown results) or also with in-sample LD matrix (not currently shown).

In new simulations, N has been considerably dropped from previous simulations. As N now varies across simulations, please cross-check once more that all methods are always run with the correct value for N.

Reviewer #3: all my concerns have been addressed

**Have all data underlying the figures and results presented in the manuscript been provided?**

Reviewer #1: Yes

Reviewer #2: Yes

Reviewer #3: Yes

PLOS authors have the option to publish the peer review history of their article (what does this mean?). If published, this will include your full peer review and any attached files.

Reviewer #1: No

Reviewer #2: No

Reviewer #3: No

---

## [Editor Report · Decision Letter 2]

17 Jun 2022

Dear Dr Carbonetto,

We are pleased to inform you that your manuscript entitled "Fine-mapping from summary data with the "Sum of Single Effects" model" has been editorially accepted for publication in PLOS Genetics. Congratulations!

Yours sincerely,

Michael P. Epstein

Associate Editor

PLOS Genetics

David Balding

Section Editor: Methods

PLOS Genetics

Comments from the reviewers (if applicable):

**Data Deposition**

http://datadryad.org/submit?journalID=pgenetics&manu=PGENETICS-D-21-01452R2

**Press Queries**

---

## [Editor Report · Acceptance letter]

14 Jul 2022

PGENETICS-D-21-01452R2 

Fine-mapping from summary data with the "Sum of Single Effects" model 

Dear Dr Carbonetto, 

We are pleased to inform you that your manuscript entitled "Fine-mapping from summary data with the "Sum of Single Effects" model" has been formally accepted for publication in PLOS Genetics! Your manuscript is now with our production department and you will be notified of the publication date in due course.

With kind regards,

Zsofia Freund

PLOS Genetics

On behalf of:
